# Breaking down the Digital Fortress: The Unseen Challenges in Healthcare Technology—Lessons Learned from 10 Years of Research

**DOI:** 10.3390/s24123780

**Published:** 2024-06-11

**Authors:** Alison Keogh, Rob Argent, Cailbhe Doherty, Ciara Duignan, Orna Fennelly, Ciaran Purcell, William Johnston, Brian Caulfield

**Affiliations:** 1Clinical Medicine, School of Medicine, Trinity College Dublin, Tallaght University Hospital, D24 TP66 Dublin, Ireland; alison.keogh@tcd.ie; 2Insight Centre for Data Analytics, University College Dublin, D04 V1W8 Dublin, Ireland; robargent@rcsi.ie (R.A.); ciara.duignan@insight-centre.org (C.D.); orna.fennelly@insight-centre.org (O.F.); ciaran.purcell@ucd.ie (C.P.); william.johnston@insight-centr.org (W.J.); b.caulfield@ucd.ie (B.C.); 3School of Pharmacy and Biomolecular Sciences, RCSI University of Medicine & Health Sciences, D02 YN77 Dublin, Ireland; 4School of Public Health, Physiotherapy and Sports Science, University College Dublin, D04 V1W8 Dublin, Ireland; 5School of Allied Health, University of Limerick, V94 T9PX Limerick, Ireland

**Keywords:** digital health, privacy, validity, reimbursement, patient involvement, interoperability

## Abstract

Healthcare is undergoing a fundamental shift in which digital health tools are becoming ubiquitous, with the promise of improved outcomes, reduced costs, and greater efficiency. Healthcare professionals, patients, and the wider public are faced with a paradox of choice regarding technologies across multiple domains. Research is continuing to look for methods and tools to further revolutionise all aspects of health from prediction, diagnosis, treatment, and monitoring. However, despite its promise, the reality of implementing digital health tools in practice, and the scalability of innovations, remains stunted. Digital health is approaching a crossroads where we need to shift our focus away from simply looking at developing new innovations to seriously considering how we overcome the barriers that currently limit its impact. This paper summarises over 10 years of digital health experiences from a group of researchers with backgrounds in physical therapy—in order to highlight and discuss some of these key lessons—in the areas of validity, patient and public involvement, privacy, reimbursement, and interoperability. Practical learnings from this collective experience across patient cohorts are leveraged to propose a list of recommendations to enable researchers to bridge the gap between the development and implementation of digital health tools.

## 1. Introduction

The proliferation of digital health technologies is generally accepted to be a revolutionary development signifying a fundamental paradigm shift in how healthcare operates [1]. Digital health is a broad term encompassing electronically captured data, along with technical and communications infrastructure and applications in the healthcare ecosystem [1]. Advances in data analytics, wearable devices, artificial intelligence, and more are packaged as solutions which will improve efficiency and connect and empower stakeholders through proactive data sharing in a timely, flexible, and integrated manner [1,2,3,4]. Commercially, technology giants such as Apple, Google, Huawei, and Samsung are adding their weight to the system, offering health and performance measurements for users to monitor themselves, a successful strategy demonstrated by their market value, which is expected to reach USD 639.4 billion by 2026 [5,6,7].

Despite its promise however, the digital health ecosystem remains murky, complex, and confusing. To date, digital health technologies have failed to demonstrate themselves as drivers of patient behaviour change [8,9,10], while sustained engagement with technologies has either been difficult to achieve, or ultimately, is not the aim, making it unclear how sustainable and scalable some solutions may be [11,12]. Furthermore, implementing digital health technologies into routine care is fragmented, owing to various systemic issues across jurisdictions. In essence, digital health is increasingly finding itself at a crossroads where it seeks to balance the continued development of innovative solutions with the real-world consequences and required adaptations necessary to actualise its potential.

Consequently, it is time for digital health researchers to pause and take stock of where we are, and where we wish to go, in this mission to improve healthcare. The authors are a group of researchers with over 10 years of experience in digital health research, specifically in the domain of the development of pre-commercial solutions, or the evaluation of technologies which are already commercially available. We have reflected on our experiences to date and identified five key lessons that we feel are currently limiting the potential for digital health technologies to develop further. In this perspective piece, we outline these lessons learned and offer recommendations for future research which we believe are fundamental to realise the potential for digital health technologies (Table 1).

## 2. Lessons Learned

### 2.1. Lesson 1: Validity Needs Revitalising to Compete with the Commercial Ecosystem

Despite the ubiquitous presence of digital health technologies, a big question remains: Can they provide valid and reliable estimations of biometric data? Validity is the foundation upon which the development of evidence-based interventions—and advancements in healthcare—are built. Despite its importance, validation poses challenges due to the dynamic nature of the sector and the distinct validation stages needed to demonstrate reliability to foster confidence in measurements [13,14].

We have undertaken validation at various stages of development [15,16,17,18,19,20,21,22,23,24,25,26,27] (Table 2). Each stage comes with its own challenges, not least the time needed to ensure each step is completed robustly [13,14,15]. This is true when developing any new hardware or software, or when independently testing existing commercial products. Consequently, traditional research dissemination methods struggle to keep pace with a commercial industry where hardware iterations are frequent and software updates, which can incorporate new processing strategies, often occur multiple times within a year. The result of this discrepancy is a lack of confidence and, thus, a potential lack of applicability for emerging technologies. Indeed, a recent umbrella review assessing the validity of consumer wearables indicated that devices show significant inaccuracies for certain metrics, particularly for the estimation of energy expenditure, step counting, and sleep and heart rate during vigorous activity [28].

Benchtop testing offers quality assurance at the basic physical unit level and can lead to the validation of higher-level measures at later stages, yet the potential for competitive advantage conflicts might deter companies from adopting this testing method. Furthermore, the demonstration of validity, and indeed the development of machine learning algorithms at this point, does not always translate to real-world validity. Nevertheless, we find the possibility of citizen science promising. Users, already equipped with devices, could contribute their data for research, strengthening validation exercises [29]. Several research institutions and companies are now embracing this approach, known as data altruism (https://shil.stanford.edu/myphd/ [accessed on 15 December 2023]; https://allofus.nih.gov/ [accessed on 15 December 2023]; https://wetrac.ucalgary.ca/ [accessed on 15 December 2023]; https://www.ukbiobank.ac.uk/ [accessed on 15 December 2023]; https://tryvital.io/ [accessed on 15 December 2023]; https://thryve.health/ [accessed on 15 December 2023]; https://www.fitabase.com/ [accessed on 15 December 2023]; https://www.labfront.com/ [accessed on 15 December 2023]; https://www.fitrockr.com/ [accessed on 15 December 2023]). However, such ‘agile’, real-world validation methods nonetheless require standardised device- and outcome-specific assessment protocols to allow pooling and comparison of data.

Herein lies the call to action for researchers: foster partnerships with research groups in other institutions and companies, and develop validation protocols that can leverage real-world data, to fast-track validation and encourage public engagement in validation studies. This collaborative effort can yield transparency, troubleshoot performance issues, and potentially offer cost savings during development. For clinicians, understand that validation is a continuous process that seeks to ensure data integrity and reliability. As digital health technologies become more intertwined with patient care, critically appraising these tools for their validity is vital to maintain patient safety and data reliability. Furthermore, there is a need for greater transparency in reporting validation methods, likely through the development of agreed standards of reporting.

### 2.2. Lesson 2: Patients Need to Be Our Partners, Not Simply Our End-Users

Irrespective of the effectiveness and validity of a technology, if the intended user is unable, or not motivated, to interact with it, it will not succeed in changing outcomes. We have gathered ample evidence that people see value in remotely gathering their health information [17,20,30,31,32,33,34,35]; however, the current reality is that monitoring may not meet expectations or may fail to answer the questions that users have [31,36,37,38,39]. True value and the focus on patient needs may get lost during the development process when the focus is typically on technical elements, while the unmet need for digital interventions is rarely considered [1,40]. We have found that usability of wearables is not formally tested or is tested in a manner that is, at best, basic in nature [30], while pilot testing of devices is rarely undertaken [37]. Thus, our experiences suggest that while many technologies are designed with patients in mind, they are not being designed with patients. This is leading to solutions which may frustrate users, which are not fit for purpose, cannot be implemented successfully, or which fail to live up to their promise.

We have extensive evidence of working alongside patients in the development of digital health technologies to monitor various conditions including knee replacement rehabilitation [17], heart failure self-management [20,35], and real-world digital mobility outcome measures [41]. We have engaged with patients across each of the domains of patient and public involvement (PPI) in our work (https://www.nihr.ac.uk/documents/briefing-notes-for-researchers-public-involvement-in-nhs-health-and-social-care-research/27371 [accessed on 15 December 2023]), mostly within the Mobilise-D consortium, a public–private partnership which has developed digital mobility outcome measures of real-world walking across multiple patient cohorts. This has led to the identification of PPI recommendations [41], changes to protocols, public facing dissemination activities, and more (https://youtu.be/qTazIpSC4DU?si=WwKYSKY2xBu2J2pe [accessed on 10 January 2024]; https://youtu.be/hherCpNiKLw?si=01E5EwPM2ww-xmc_ [accessed on 1 June 2024]; https://youtu.be/3FwD9XZynHo?si=nHSkxjqQpQxGJDBp [accessed on 15 December 2023]; https://youtu.be/Y_rfqCROIDQ?si=pRI2Fq0O49Bm5B4M [accessed on 15 December 2023]; https://mobilise-d.eu/ppag-activities-and-contributions/ [accessed on 15 December 2023]). Engaging meaningfully in this user-centred design approach means that the solution might not eventually be the one that was originally envisaged, it might not be a net positive for all types of users, nor might it be simply a digitisation of the current care pathway. However, understanding fundamental needs and barriers and facilitators to solutions may enable better engagement and impact and will certainly result in a reduction in waste.

We consider the continuing lack of PPI to be a significant barrier to the successful implementation of digital health technologies. Funding and regulatory bodies are beginning to acknowledge this by making PPI mandatory in submissions from academia and industry. Consequently, it is imperative that researchers and clinicians include it as standard in their work. We encourage researchers and clinicians to engage with the various bodies and organisations that now exist to support researchers with this. This includes PPI guidelines, patient societies, and bodies who support and train patients to be research partners (https://eupati.eu/ [accessed on 10 January 2024]; https://ipposi.ie/ [accessed on 10 January 2024]) as well as academic institutional supports to support PPI and design thinking training (e.g., https://ppinetwork.ie/ [accessed on 10 January 2024]). Finally, we call on researchers and clinicians to actively challenge industry partners and start-ups about how PPI has been integrated into the design of their solutions prior to implementing them in studies or practice.

### 2.3. Lesson 3: Digital Health’s Double-Edged Sword—Innovation vs. Privacy

Continuous health monitoring brings with it immense potential but also significant threats to confidentiality and privacy, particularly with the increasing volume of commercial tools. The trajectory of this rapidly evolving ecosystem, swayed by the influence of regulatory architecture, could culminate in either utopian or dystopian outcomes. The former envisages an environment characterised by comprehensive regulations guaranteeing judicious data utilisation for maximum societal benefit, with informed consent and privacy enshrined as fundamental tenets [42,43]. In contrast, the dystopian scenario foreshadows an arena of rampant misuse by healthcare data brokers, unauthorised data access, privacy transgressions, and the commodification of health data, catalyzed by inadequate regulations and discordant international standards [42]. The reality is possibly somewhere in the middle. Technical infrastructure can support the secure sharing of data in locked environments which retain privacy. Currently, we are exploring work in relation to the regulatory and technical safeguards required to do this within Ireland, in a way that supports federated data sharing within the European Union [44,45].

Our interactions with patients suggest that their behavior relates to the trust they have in the person or institution implementing the technology [46]. Specifically, there is an assumption that researchers and healthcare professionals have participants’ best interests at heart and are unlikely to engage with technologies that may put them at risk. However, the privacy paradox has shown that despite concerns, people readily disclose and share data with various companies, including those they have low trust in [47,48,49]. This paradox is domain dependent and is linked to technical literacy [47,48,50], emphasising the great responsibility that falls upon researchers as the gatekeepers of participant’s privacy.

However, within this, we have experienced another conflict that limits the potential for researchers to progress the digital health space. Specifically, in some jurisdictions, although commercial products are available for individuals to use and purchase, when researchers seek to evaluate these same products in studies, they are met with walls of academic institutional privacy barriers. This includes the need for data processing agreements with the companies whose products are being used. Some companies do not wish to formally engage with researchers seeking to independently assess their products and therefore will not enter into data processing agreements with them. Others simply do not see the need for it as their commercially available products are not intended as research tools. Thus, while we acknowledge the importance of thorough data management procedures, we must nonetheless admit that these standards are also limiting the independent testing of existing digital health technologies and consequently reduce our ability to evaluate their effectiveness, validity, and implementation.

In light of this, we propose that all biometric data be considered digital specimens, warranting the same rigor, care, and caution accorded to their physical analogues [43]. Privacy, in this context, transcends the basic need for data protection to encompass the individual’s right to dictate the access, manipulation, and dissemination of their personal data. In the commercial space, the urgent necessity for privacy is underscored by the potential misappropriation by health data brokers and the shortcomings of end-user licence agreements, which tend to prioritise corporate immunity over user protection [43,49]. Consequently, it is incumbent upon researchers and clinicians to adopt a cautious and informed approach when interpreting data from consumer devices presented by patients, considering the source and validity of the data (see lesson 1), while providing counsel on data privacy and protective measures. Finally, in order to ensure that privacy concerns are holistically addressed and allayed, healthcare professionals are urged to engage in a detailed and systematic evaluation of each device’s security measures (Table 3).

In conjunction with this, security issues need to be addressed alongside privacy. Digital health researchers might not possess the specialised skills necessary to thoroughly evaluate the security of digital tools. Recognising this, it is important to highlight existing security standards and processes that healthcare systems and research organisations are adopting to address these challenges. For instance, the Digital Technology Assessment Criteria (DTAC) in the UK provides a standardised framework to assess the security and clinical safety of digital health technologies. Similarly, ORCHA (Organisation for the Review of Care and Health Apps) conducts evaluations of health apps worldwide to ensure they meet predefined security and privacy standards. Although these processes have their limitations and challenges, they represent significant steps toward systematising security assessments in digital health. At the institutional level, many research organisations and universities have established security review protocols and requirements. These internal reviews are important for ensuring that digital health technologies used in research comply with necessary security standards, thus mitigating risks associated with data breaches and unauthorised access.

### 2.4. Lesson 4: The Interplay of Commercialisation and Reimbursement in Shaping Digital Health’s Real-World Reach

Widespread adoption of digital health technologies requires a business model that is suitable for all stakeholders, a method of reimbursement that sustains their development and implementation beyond the life of a research grant. A number of major players who have attracted investment in recent years have pivoted from their initial offering to provide a sustainable business model which delivers a new care pathway, rather than offering a technology to be embedded within an existing one (e.g., https://www.hingehealth.com [accessed on 15 December 2023]; https://swordhealth.com [accessed on 15 December 2023]). Whilst these pivots have been innovative, and potentially more disruptive than their initial offering, they were largely driven by the need for a sustainable reimbursement model that provides cost-effectiveness for all.

Despite positive outcomes in early research stages, projects often fail to achieve adoption as they are not financially sustainable. This is certainly the experience of the authors, who have explored commercial opportunities of research outputs in the domain of physical therapy and found the primary stumbling block to be the prevalence of fee-for-service models, a barrier that has been previously highlighted elsewhere [51]. In many cases, where digital health technologies can lead to proactive and preventative healthcare management, they seek to reduce the utilisation of services or contact points in the system. For many using a fee-for-service model, increasing expenditure for a tool which improves efficiency but reduces the number of clinic visits, and therefore income, is counterintuitive. Whilst the technology may provide better patient care, there is no motivation for the buyer to adopt the system into practice.

Conversely, in public health systems, the motivation to improve efficiency can lead to cost savings. The challenge for achieving implementation though is in gathering the required evidence to prove cost-effectiveness, which can take many years and, as such, requires a large amount of upfront investment and associated risk. Consequently, in the author’s experiences, the first point of entry of new technologies is rarely public health systems. As a result, rather than revolutionising healthcare, digital health technologies developed in fee-for-service models actually risk increasing health inequality and the digital divide, rather than reducing it [52,53,54,55,56]. There are growing moves towards bundled payment models in the form of value-based care [51,57], where payment is based on the outcome of care, rather than the quantity, thus providing motivation to offer the most efficient service whilst still delivering high-quality care. There is a need for regulators, national governments, departments of health, etc., to become more closely invovled in planning for such shifts in policy to effect meaningful change.

We consider the use of appropriate reimbursement models to be critical to facilitate the adoption of digital health technologies, and we recommend all researchers and clinicians consider cost-effectiveness when designing or selecting a digital health technology to implement [58]. Researchers should consider the variety of reimbursement mechanisms, how they differ between jurisdictions, and the evidence requirements associated with each. All stakeholders can actively lobby authorities to adapt their reimbursement mechanisms to embrace the opportunity of digital health, whether it is value-based care or the successful DiGA framework in Germany [59], which allows for the prescription and payment of digital health interventions to be funded much like pharmacological interventions.

### 2.5. Lesson 5: Digital Health’s Future Hinges on Interoperability

The true potential for innovation lies in the realm of interoperability. Serving as the fundamental cornerstone for effectively harnessing digital health data, the aim of interoperability is to bridge the chasm that exists between insular data repositories and individual health technologies. As it stands however, the current digital health landscape is more reminiscent of a mosaic of disjointed ‘small data’, as opposed to the idealised concept of ‘big data’. Indeed, our own research projects have highlighted the barriers that exist to the adoption and usefulness of digital health technology, as a result of siloed information that is difficult for anyone other than end-users to access or act upon [17,33,34,35,36,37]. Furthermore, proprietary systems typically fail to, and are not required to, provide easy access to third parties, thus limiting data flows and innovation. Interoperability can be both syntactic, whereby systems cannot communicate with each other, or semantic, where even if we get access to the data, it is in different formats, which preclude aggregation [44]. Consequently, for digital health to realise its full potential, there is a need to design technologies that facilitate and provide seamless communication across IT systems. Linked to this is the need to establish both standardised data formats internationally [60] and promote consistent use of standardised terminologies such as SNOMED-CT, DICOM, and LOINC where possible [60,61].

In short, drawing from all lessons, the cost-effectiveness and successful implementation of digital health technologies requires a fundamental shift for researchers to solve this current lack of interoperability [62]. Beyond technical requirements, there are many other organisational requirements that are also needed to ensure interoperability. Currently, researchers seek to design tools that are effective, valid, and useful, and then wait to consider where they fit within care pathways, who pays for them, and how they operate within a system. A pivot towards considering interoperability early in the process is crucial for the widespread adoption of digital health technologies, as well as for the general advancement of medical research. Further, interoperability has the capacity to enhance the overall quality of research, as data can be scrutinised by experts globally and across a myriad of sources.

Therefore, interoperability may well hold the key to unlocking the viability of digital health technologies by bolstering their cost-effectiveness and amplifying their capacity to deliver high-quality care. Not-for-profit organisations such as openEHR provide open specifications for the management, storage, and retrieval of data in electronic health records, while international standards for data structure such as Health Level Seven International (HL7), Fast Healthcare Interoperability Resources (FHIR), and Integrating the Healthcare Enterprise (IHE) now exist. Furthermore, the European Health Data Space and other federated data analysis projects (e.g., European Open Science Cloud: https://research-and-innovation.ec.europa.eu/strategy/strategy-2020-2024/our-digital-future/open-science/european-open-science-cloud-eosc_en [accessed on 10 January 2024]) promote the use of these standards which will become mandatory future requirements. We therefore recommend that researchers invest early in backend development to future-proof their infrastructure to be scalable, to have open APIs, and to consider their use beyond their own projects. Once the technology and data are there, we need unique identifiers to link people across datasets, along with incentives and legislation to ensure that sharing occurs and that there is security of the access methods to the data [63].

## 3. Conclusions

This paper has summarised the collective experiences of a group of digital health researchers to highlight continued barriers and considerations in the effectiveness and implementation of digital health technologies. When standing at a crossroads, we have a choice, continue on as we are or change direction. If digital health research continues on its current path, it risks a never-ending cycle of unfulfilled potential, development without implementation, an on-going conflict between researchers and commercial entities with the patient caught in the middle, and the delivery of fragmented care which increases health inequities and the digital divide. Digital health is a complex, messy, and multi-faceted domain, and targeted changes in the way we conduct research are needed to move us forward. We do not propose to have all the answers; however, we have sought outline key recommendations in the areas of validity, patient and public involvement, cost-effectiveness, privacy, and interoperability, based on our lessons learned, as a call to action for future studies and solution development in this space to implement and make meaningful change to healthcare outcomes.

## Figures and Tables

**Table 1 sensors-24-03780-t001:** List of recommendations for digital health researchers for future digital health projects.

	Recommendation	Lesson Linked to
1	Leverage real-world data to develop new validation protocols.	Validity
2	Foster partnerships with companies and research groups looking to use citizen science, real-world validation.	
3	Encourage public involvement in validation studies.	
4	Ensure validity at multiple time points has been measured before implementing a tool clinically.	
5	Engage with existing academic structures that can support the development and integration of PPI into studies from the start.	Patient and public involvement
6	Adopt a user-centred design process, with PPI contributors as equal partners as standard within studies.	
7	Actively challenge industry partners/startups about how PPI has been integrated into their solutions, prior to implementing them.	
8	Cautiously interpret data from commercial devices considering the source and validity of the data.	Data privacy
9	Counsel participants and patients on data privacy and protection measures.	
10	Engage in a detailed and systematic evaluation of each device’s security measures prior to implementation.	
11	Consider reimbursement models during the design process—who will pay for it, how does it fit within current models, or are new models and pathways needed?	Cost-effectiveness
12	Consider cost-effectiveness before implementing tools in studies.	
13	Lobby authorities to adapt their reimbursement mechanisms to embrace the opportunity of digital health.	
14	Consider interoperability early in the design process—what is needed to allow the tool to integrate with existing pathways and other tools?	Interoperability
15	Invest in backend capabilities early that will allow infrastructure to be scalable.	
16	Have open APIs within tools and consider their use beyond their own projects from the start.	

**Table 2 sensors-24-03780-t002:** Device evaluation stages based on the work of Keadle et al. and Ash et al. [14,15].

Validity Stage				
	Benchtop	Laboratory	Free-Living	Implementation
Aim of stage	The device is evaluated in response to standardised synthetic signals.	The device is tested in human participants under controlled conditions; outputs are compared to gold standard criterion measures.	The device is tested in human participants in naturalistic and variable (‘free-living’) conditions; outputs are compared to field-based or practical criterion measures.	The device is utilised in a healthcare research setting, where its performance, usability, and impact on patient outcomes are evaluated.
Example process for stage (based on accelerometer to measure step counts)	Attach accelerometer to calibrated shaker plate and compare its outputs to the expected accelerations.	Participants undergo a standardised walking test wearing the device, and the results are compared with gold standard tests (i.e., motion capture cameras).	Participants wear the device during daily activities, and device-measured step count is compared with another validated device.	The device is used in a clinical trial to monitor patient step counts remotely. Its ability to accurately capture data, its ease of use for patients and staff, and its impact on patient outcomes are assessed.

**Table 3 sensors-24-03780-t003:** Steps for researchers to consider when evaluating privacy and security concerns with digital health technologies.

	Steps and Questions to Consider
1	Does the company have a privacy policy that clearly outlines how they collect, use, and store personal data?
2	Are there controls in place to prevent unauthorised access to personal data, such as strong passwords and secure login procedures?
3	Does the device have physical security measures in place, such as a secure enclosure or tamper-resistant hardware?
4	Is personal data encrypted when it is transmitted or stored on the device or on the company’s servers?
5	Does the company have a process in place for responding to data breaches or other security incidents?
6	Can users opt out of data collection or delete their personal data if they choose to do so?
7	Can users control the data that is collected and shared by the device, such as by adjusting privacy settings or disabling certain features?
8	Are there clear terms of service that explain how personal data may be used, including any third-party data sharing?
9	Are there physical security measures in place to protect personal data, such as secure servers and data centres?
10	Is the company transparent about any third-party data sharing or data analytics that may be conducted with personal data?
11	Does the company have clear processes for obtaining informed consent from users before collecting or using their personal data?
12	Is the company compliant with relevant privacy laws and regulations, such as the General Data Protection Regulation (GDPR) in the European Union and the California Consumer Privacy Act (CCPA) in the United States?
13	Who has control over the data that is generated using a digital health tool? Are there adequate controller–processor agreements in place if required by law?

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
