# Peer review of "Breaking down the Digital Fortress: The Unseen Challenges in Healthcare Technology—Lessons Learned from 10 Years of Research"

_sensors, 2024, doi:10.3390/s24123780_

Round 1

Reviewer 1 Report

Comments and Suggestions for Authors

The authors have provided an evidence-based explanation of challenges to incorporating devices in healthcare and lessons learned in developing and implementing these devices. 

The piece is clear, concise, and easily readable. Tables included are very helpful in detailing the basis for lessons learned and clarifying with examples. As a reader, I come away with a very clear understanding of the authors' recommendations.

Below are just a few points of clarification and revision to address: 

Lines 23-25: This sentence makes me think you are doing some kind of literature review that summarizes results. I would revise, maybe combining with the following sentence to be clear that you are drawing from 10 years of experience, but not summarizing the entirety of your work, since that is not your purpose.

Line 65: The title contains the word Eommercial—is this a typo?

Line 123: Extra comma after “or”

Lines 129-30: Page not found via link provided

Lines 131-35: Please provide context for the youtube videos—What is Mobilise-D? A very brief explanation of what the group is would be helpful to the reader.  

Lines 288-290: Page not found via link provided 

Author Response

Reviewer 1

The authors have provided an evidence-based explanation of challenges to incorporating devices in healthcare and lessons learned in developing and implementing these devices.

The piece is clear, concise, and easily readable. Tables included are very helpful in detailing the basis for lessons learned and clarifying with examples. As a reader, I come away with a very clear understanding of the authors' recommendations.

Authors: Many thanks for your review and these insights.

Below are just a few points of clarification and revision to address:

Lines 23-25: This sentence makes me think you are doing some kind of literature review that summarizes results. I would revise, maybe combining with the following sentence to be clear that you are drawing from 10 years of experience, but not summarizing the entirety of your work, since that is not your purpose.

Authors: We have altered this to highlight that the work is based on our experiences rather than a review.

Line 65: The title contains the word Eommercial—is this a typo?

Authors: We have adjusted this typo

Line 123: Extra comma after “or”

Authors: Completed

Lines 129-30: Page not found via link provided

Authors: Link updated

Lines 131-35: Please provide context for the youtube videos—What is Mobilise-D? A very brief explanation of what the group is would be helpful to the reader.

Authors: We have provided a brief explanation of what the Mobilise-D consortium is.

Lines 288-290: Page not found via link provided

Authors: Link updated

Reviewer 2 Report

Comments and Suggestions for Authors

This is a really interesting and useful summary of lessons learned and recommendations for the future of digital health research. The breadth and depth of knowledge of the research team in this area is well evidenced throughout. The areas focused on certainly pose significant challenges for the future of the field.

It was very encouraging to see the focus on PPI throughout and the careful  consideration of the challenges and progress in negotiating PPI in digital health.  

I really enjoyed reading the review and look forward to seeing it in print.

Some minor suggestions to contextualise some elements and enhance the paper:

- The privacy steps/questions are useful. However, it would be good to acknowledge security standards and processes in digital health here. Many digital health researchers will not be well-placed to evaluate the security of digital tools as this can be highly specialist work. Rather than relying on individual researchers assessing this, it would be good to note how healthcare systems are already moving in this direction with their own processes / requirements and how a system-level effort might be required (e.g. DTAC in the UK; ORCHA processes worldwide - albeit each of these with their own limitations and challenges). At a research organisation level / university, many organisations will have security reviews and requirements in place too. Some of acknowledgment of this would be useful.

- The point about sustainability and reimbursement models is well made. While encouraging digital health researchers to lobby authorities may be useful, individual researchers are often quite powerless to shape and transform the healthcare delivery models. Even developing elaborate cost-effectiveness models can be powerless if there is no system in place to fund digital health tools. It may be useful to mention here that the authorities  - depts of health / regulators  etc need to get involved more closely with this and come on board for real change to happen.

- It was great to see the focus on interoperability. It would be useful to note that this is often restricted by locked-down proprietary systems (e.g. EHR providers) that are often not required to provide easy access to digital health third parties, thereby restricting data flows and limiting innovation. 

Author Response

This is a really interesting and useful summary of lessons learned and recommendations for the future of digital health research. The breadth and depth of knowledge of the research team in this area is well evidenced throughout. The areas focused on certainly pose significant challenges for the future of the field. It was very encouraging to see the focus on PPI throughout and the careful consideration of the challenges and progress in negotiating PPI in digital health. I really enjoyed reading the review and look forward to seeing it in print.

Authors: Many thanks for your review and these insights.

Some minor suggestions to contextualise some elements and enhance the paper:

- The privacy steps/questions are useful. However, it would be good to acknowledge security standards and processes in digital health here. Many digital health researchers will not be well-placed to evaluate the security of digital tools as this can be highly specialist work. Rather than relying on individual researchers assessing this, it would be good to note how healthcare systems are already moving in this direction with their own processes / requirements and how a system-level effort might be required (e.g. DTAC in the UK; ORCHA processes worldwide - albeit each of these with their own limitations and challenges). At a research organisation level / university, many organisations will have security reviews and requirements in place too. Some of acknowledgment of this would be useful.

Authors: Thank you for your input. We have made the following changes to address your comments:

  1. Acknowledgment of Security Standards and Processes:

We have added a new section discussing the importance of security alongside privacy in digital health. This includes acknowledging that digital health researchers may not have specialized skills to evaluate the security of digital tools. We have also highlighted existing security standards and processes, including the Digital Technology Assessment Criteria DTAC and Organisation for the Review of Care and Health App) processes worldwide.

  1. System-Level Efforts:

We now emphasized the role of system-level efforts and institutional security review protocols in ensuring robust security measures.

  1. Integration of Institutional and System-Level Security Reviews:

 We recommend that researchers leverage existing frameworks and institutional resources to enhance the robustness of security assessments.

  1. Detailed Steps for Evaluating Privacy and Security Concerns:

 We retained and slightly rephrased the detailed steps for researchers to consider when evaluating privacy and security concerns, ensuring they align with the new context of leveraging existing security standards and institutional reviews.

- The point about sustainability and reimbursement models is well made. While encouraging digital health researchers to lobby authorities may be useful, individual researchers are often quite powerless to shape and transform the healthcare delivery models. Even developing elaborate cost-effectiveness models can be powerless if there is no system in place to fund digital health tools. It may be useful to mention here that the authorities - depts of health / regulators etc need to get involved more closely with this and come on board for real change to happen.

Authors: We have added in a line to emphasise this need.

- It was great to see the focus on interoperability. It would be useful to note that this is often restricted by locked-down proprietary systems (e.g. EHR providers) that are often not required to provide easy access to digital health third parties, thereby restricting data flows and limiting innovation.

Authors: We have added in a line to highlight this barrier.